# Induction chemotherapy followed by definitive chemoradiotherapy versus chemoradiotherapy alone in esophageal squamous cell carcinoma: a randomized phase II trial

Shiliang Liu [1,2,5], Liling Luo [1,3,5], Lei Zhao [1,2,5], Yujia Zhu [1,2,5], Hui Liu[1,2], Qiaoqiao Li[1,2], Ling Cai[1,2], Yonghong Hu[1,2], Bo Qiu [1,2], Li Zhang [1,2], Jingxian Shen [1,4], Yadi Yang [1,4], Mengzhong Liu [1,2,6✉] & Mian Xi [1,2,6✉]

This randomized phase II trial aims to compare the efficacy and safety of induction chemotherapy followed by definitive chemoradiotherapy (CRT) versus CRT alone in patients with esophageal squamous cell carcinoma (ESCC) unsuitable for surgery ($N = 110$). The primary outcome was overall response rate (ORR), whereas the secondary outcome was overall survival. This trial did not meet pre-specified endpoints. The ORR was 74.5% in the induction chemotherapy group versus 61.8% in the CRT alone group ($P = 0.152$). The 3-year overall survival rate was 41.8% in the induction chemotherapy group and 38.1% in the CRT alone group ($P = 0.584$; hazard ratio, 0.88; 95% CI, 0.54–1.41). Grade 3–5 adverse events were similar. Patients who responded to induction chemotherapy had improved survival in the post-hoc analysis. These results demonstrate no improvement in response rate or survival with the addition of induction chemotherapy to CRT in unselected patients with ESCC. Trial number: NCT02403531.

[1] State Key Laboratory of Oncology in South China, Collaborative Innovation Centre for Cancer Medicine, Guangdong Esophageal Cancer Institute, Guangzhou, China. [2] Department of Radiation Oncology, Sun Yat-sen University Cancer Center, Guangzhou, China. [3] Department of Radiation Oncology, Guangdong Provincial People's Hospital, Guangdong Academy of Medical Sciences, School of Medicine, South China University of Technology, Guangzhou, China. [4] Department of Imaging Diagnosis and Interventional Center, Sun Yat-sen University Cancer Center, Guangzhou, China. [5] These authors contributed equally: Shiliang Liu, Liling Luo, Lei Zhao, Yujia Zhu. [6] These authors jointly supervised this work: Mengzhong Liu, Mian Xi.
✉email: liumzh@sysucc.org.cn; ximian@sysucc.org.cn

As the sixth leading cause of cancer deaths worldwide, esophageal cancer (EC) is a lethal disease with an unsatisfactory prognosis[1]. Concurrent chemoradiotherapy (CRT) is the standard of care for patients with EC unsuitable for surgery[2]. Despite therapeutic advances in radiotherapy modality and chemotherapy regimens in recent years, the prognosis for patients who receive definitive CRT remains unfavorable, with 3-year overall survival (OS) rates of 26.9–55.4%[3–8]. More importantly, over 50% of patients develop locoregional or distant recurrences after definitive CRT[9,10]. Therefore, more effective treatment regimens are greatly needed.

With the potential for improvement in locoregional control, early eradication of micrometastases, and supporting organ preservation, the addition of induction chemotherapy prior to concurrent CRT is a theoretically attractive approach in EC. Several retrospective studies have suggested that the combination of induction chemotherapy and CRT could improve tumor response as well as survival outcomes[11,12]. This strategy has also been investigated by prospective, uncontrolled phase II trials, indicating promising efficacy and manageable toxicities, especially in localized esophageal squamous cell carcinoma (ESCC)[13–15]. However, given the lack of high-quality data from prospective randomized trials, the value of induction chemotherapy before definitive CRT has not yet been established in EC.

Here, we conducted a prospective, randomized phase II trial to compare the efficacy and safety of induction chemotherapy followed by definitive CRT vs. CRT alone in patients with ESCC unsuitable for surgery. Additionally, we performed an exploratory, post-hoc analysis to analyze the relationship between tumor response to induction chemotherapy and treatment outcomes.

## Results

**Patient characteristics**. From May 2015 to September 2017, 126 patients were assessed and 110 were randomly assigned to the two treatment groups: 55 to the IC + CRT group and 55 to the CRT group (Fig. 1). Since all patients started protocol-defined intervention, all of them were included in the safety population. Patient and tumor characteristics at baseline were well balanced between the two groups (Table 1 and Supplementary Data 1).

**Treatment compliance**. All patients in the IC + CRT group completed the scheduled two cycles of induction chemotherapy; 7 of 55 patients (12.7%) had dose reductions of docetaxel or cisplatin, mainly due to hematological toxicities (in 5 patients). The delays of induction chemotherapy lasting more than 3 days occurred in 10 patients (18.2%), owing to adverse events or other reasons (Supplementary Table 1).

In the IC + CRT group, the cumulative proportions of patients who completed at least 2, 3, 4, and 5 weeks of concurrent chemotherapy during radiotherapy were 100.0%, 90.9%, 83.6%, and 65.5%, respectively. The corresponding numbers for the CRT group were 100.0%, 98.2%, 87.3%, and 80.0%, respectively (Supplementary Fig. 1A). The proportions of patients who completed concurrent chemotherapy at different points were similar in the two groups. Radiotherapy compliance for the two groups is detailed in Supplementary Table 2 and Supplementary Fig. 1B. A total of 51 patients (92.7%) in the IC + CRT group vs. 53 patients (96.4%) in the CRT group received at least 50 Gy of radiation ($P = 0.679$).

**Tumor response**. Tumor responses after induction chemotherapy and CRT are listed in Table 2. Overall, 45.5% (95% confidence interval [CI], 31.9–59.0) of the patients (25 of 55) had a response after induction chemotherapy and were defined as responders; no patients had a complete response, and 25 had a partial response. At 3 months after CRT, 74.5% (95% CI, 62.7–86.4) of the patients in the IC + CRT group (26 complete response and 15 partial response) vs. 61.8% (95% CI, 48.6–75.1) of the patients in the CRT group (22 complete response and 12 partial response) achieved a response ($P = 0.152$).

**Survival**. At the last follow-up on September 10, 2020, the median follow-up was 24.8 months (range, 2.4–63.9 months) for the whole cohort and 43.2 months (range, 15.4–63.9 months) for survivors. We recorded a total of 77 events of death or recurrence in the overall population, including 39 (70.9%) events in the IC + CRT group and 38 (69.1%) in the CRT group. Details regarding the reasons for death are provided in Supplementary Table 3. No significant differences were observed in the cumulative incidences of recurrences between the two groups ($P = 0.795$; Supplementary Fig. 2). Of the 20 patients

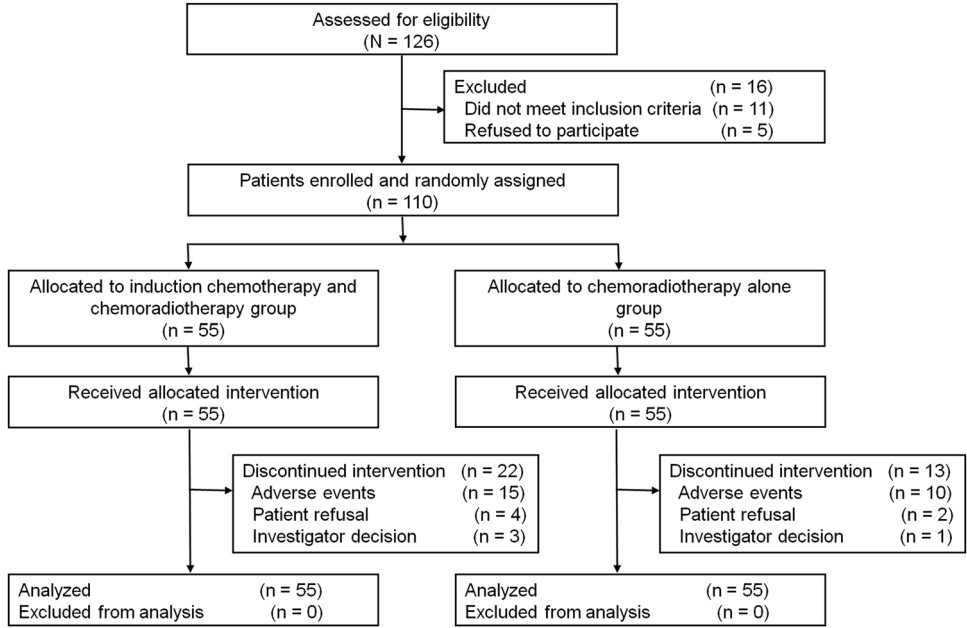

**Fig. 1 CONSORT diagram of patient flow.** This figure shows reasons for exclusion from the study and the numbers of patients included in the analyses.

## Table 1 Patient characteristics.

| Characteristic | IC + CRT group (n = 55), % | CRT group (n = 55), % |
|---|---|---|
| **Age (years)** | | |
| Median (IQR) | 57 (52–62) | 60 (54–65) |
| ≤58 | 31 (56.4) | 25 (45.5) |
| >58 | 24 (43.6) | 30 (54.5) |
| **Sex** | | |
| Male | 39 (70.9) | 43 (78.2) |
| Female | 16 (29.1) | 12 (21.8) |
| **Smoking history** | | |
| Yes | 40 (72.7) | 41 (74.5) |
| No | 15 (27.3) | 14 (25.5) |
| **Alcohol history** | | |
| Yes | 32 (58.2) | 36 (65.5) |
| No | 23 (41.8) | 19 (34.5) |
| **Weight loss within 3 months** | | |
| <10% | 44 (80.0) | 43 (78.2) |
| ≥10% | 11 (20.0) | 12 (21.8) |
| **BMI (kg/m²)** | | |
| Median (IQR) | 21.6 (19.9–23.3) | 20.3 (18.6–22.0) |
| ≤18.5 | 5 (9.1) | 12 (21.8) |
| >18.5 | 50 (90.9) | 43 (78.2) |
| **ECOG performance status** | | |
| 0 | 28 (50.9) | 21 (38.2) |
| 1–2 | 27 (49.1) | 34 (61.8) |
| **Histologic grade** | | |
| Gx/G1/G2 | 47 (85.5) | 49 (89.1) |
| G3 | 8 (14.5) | 6 (10.9) |
| **Tumor location** | | |
| Upper | 32 (58.2) | 26 (47.3) |
| Middle | 20 (36.4) | 23 (41.8) |
| Distal | 3 (5.5) | 6 (10.9) |
| **Primary tumor length (cm)** | | |
| Median (IQR) | 5 (4–7) | 5 (4–7) |
| ≤5 | 23 (41.8) | 27 (49.1) |
| >5 | 32 (58.2) | 28 (50.9) |
| **Clinical T stage** | | |
| T1–2 | 16 (29.1) | 15 (27.3) |
| T3–4 | 39 (70.9) | 40 (72.7) |
| **Clinical N stage** | | |
| N0 | 7 (12.7) | 9 (16.4) |
| N1 | 48 (87.3) | 46 (83.6) |
| **Clinical M stage** | | |
| M0 | 40 (72.7) | 36 (65.5) |
| M1a | 15 (27.3) | 19 (34.5) |
| **Clinical TNM stage** | | |
| IIA | 7 (12.7) | 8 (14.5) |
| IIB | 8 (14.5) | 9 (16.4) |
| III | 25 (45.5) | 19 (34.5) |
| IVA | 15 (27.3) | 19 (34.5) |
| **Reason for no surgery** | | |
| Inoperable | 40 (72.7) | 38 (69.1) |
| Surgical contraindication | 9 (16.4) | 11 (20.0) |
| Patient refusal | 6 (10.9) | 5 (9.1) |
| **Radiotherapy modality** | | |
| 3DCRT | 2 (3.6) | 4 (7.3) |
| IMRT | 53 (96.4) | 51 (92.7) |

IC induction chemotherapy, CRT chemoradiotherapy, IQR interquartile range, BMI body mass index, ECOG Eastern Cooperative Oncology Group, 3DCRT three-dimensional conformal radiotherapy, IMRT intensity-modulated radiotherapy.

## Table 2 Tumor response to treatment.

| Response | IC + CRT group (n = 55), % | CRT group (n = 55), % |
|---|---|---|
| **After IC** | | |
| CR | 0 (0.0) | |
| PR | 25 (45.5) | |
| SD | 28 (50.9) | |
| PD | 2 (3.6) | |
| **After CRT** | | |
| CR | 26 (47.3) | 22 (40.0) |
| PR | 15 (27.3) | 12 (21.8) |
| SD | 5 (9.1) | 5 (9.1) |
| PD | 8 (14.5) | 12 (21.8) |
| Not evaluable | 1 (1.8) | 4 (7.3) |

IC induction chemotherapy, CRT chemoradiotherapy, CR complete response, PR partial response, SD stable disease, PD progressive disease.

The 3-year OS rate was 41.8% (95% CI, 28.8–54.4) in the IC + CRT group compared to 38.1% (95% CI, 25.1–51.0) in the CRT group ($P = 0.584$; hazard ratio [HR], 0.88; 95% CI, 0.54–1.41; Fig. 2a). Similarly, no significant differences were identified in 3-year PFS between the IC + CRT and CRT groups (30.6% [95% CI, 19.0–43.0] vs. 29.8% [95% CI, 18.2–42.3], $P = 0.770$; HR, 0.94; 95% CI, 0.60–1.46; Fig. 2b).

**Exploratory analysis.** To assess the consistency of treatment effect on OS, post-hoc subgroup analyses according to the baseline characteristics are shown in Fig. 3. The interaction analysis revealed a non-significant interaction effect across all subgroups except for the clinical TNM stage subgroups ($P = 0.017$). Kaplan–Meier analysis of OS in the two treatment groups stratified by clinical TNM stage is shown in Fig. 4. For patients with stage III/IVA ESCC, the IC + CRT group had better OS than the CRT group but without statistical significance ($P = 0.069$; HR, 0.60; 95% CI, 0.35–1.04). Moreover, the IC + CRT group showed inferior OS in patients with stage II ESCC without a statistical difference ($P = 0.058$; HR, 2.85; 95% CI, 0.97–8.40).

The potential effect of tumor response to induction chemotherapy on survival outcomes was also analyzed as an exploratory, post-hoc analysis. The overall response rate (ORR) after CRT was 96.0% (95% CI, 87.7–100.0) in the responders (24 of 25) and 56.7% (95% CI, 37.8–75.5) in the nonresponders (17 of 30), respectively ($P = 0.001$). As shown in Fig. 5, the responders to induction chemotherapy had significantly more favorable survival compared with nonresponders, or with patients in the CRT group, with corresponding 3-year OS rates of 80.0% (95% CI, 58.4–91.2), 10.0% (95% CI, 2.6–23.6), and 38.1% (95% CI, 25.1–51.0), and 3-year PFS rate of 55.3% (95% CI, 33.8–72.3), 10.0% (95% CI, 2.6–23.6), and 29.8% (95% CI, 18.2–42.3), respectively ($P < 0.001$ for OS and PFS). Additionally, both OS and PFS were better in the CRT group than in the nonresponders ($P = 0.009$ for OS; $P = 0.019$ for PFS).

**Adverse events.** During induction chemotherapy, 10 of 55 patients (18.2%) had acute adverse events of grade 3 or 4, and no grade 5 toxicity occurred (Supplementary Table 4). Although 16.4% of patients (9 of 55) developed G3/G4 neutropenia after induction chemotherapy, the incidence of febrile neutropenia was only 3.6% (2 of 55), which was not serious.

Over the entire treatment phase, 21 patients (38.2%) in the IC + CRT group and 19 patients (34.5%) in the CRT group had grade 3–5 adverse events ($P = 0.692$). The IC + CRT group had a higher incidence of grade 3 or 4 neutropenia than the CRT group, but

with locoregional recurrence only (10 in the IC + CRT group and 10 in the CRT group), 8 patients underwent salvage surgery, 3 received salvage CRT, 5 received palliative chemotherapy, and 4 received supportive care due to poor performance status.

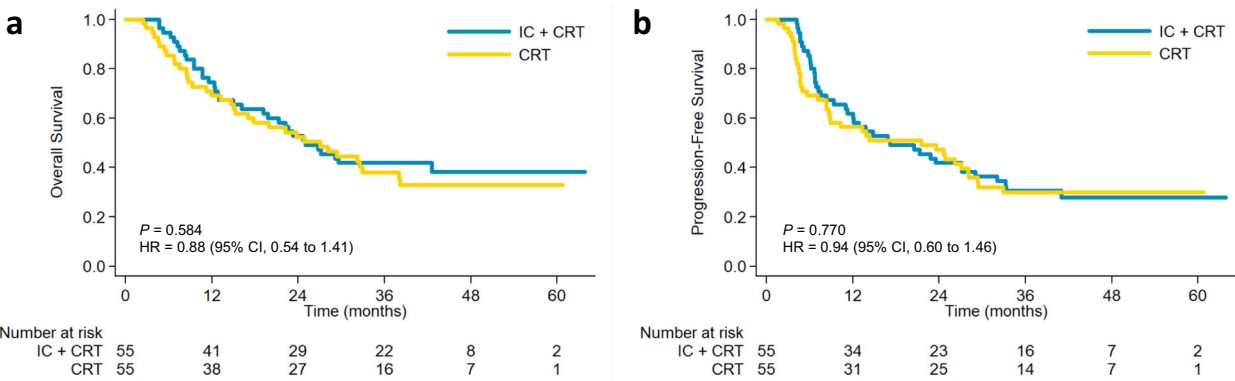

**Fig. 2 Kaplan–Meier estimates of survival curves for the two treatment groups. a** Overall survival; **b** progression-free survival. IC + CRT induction chemotherapy followed by concurrent chemoradiotherapy, CRT chemoradiotherapy alone. Log-rank test was used (2-sided). Source data are provided as a Source Data file.

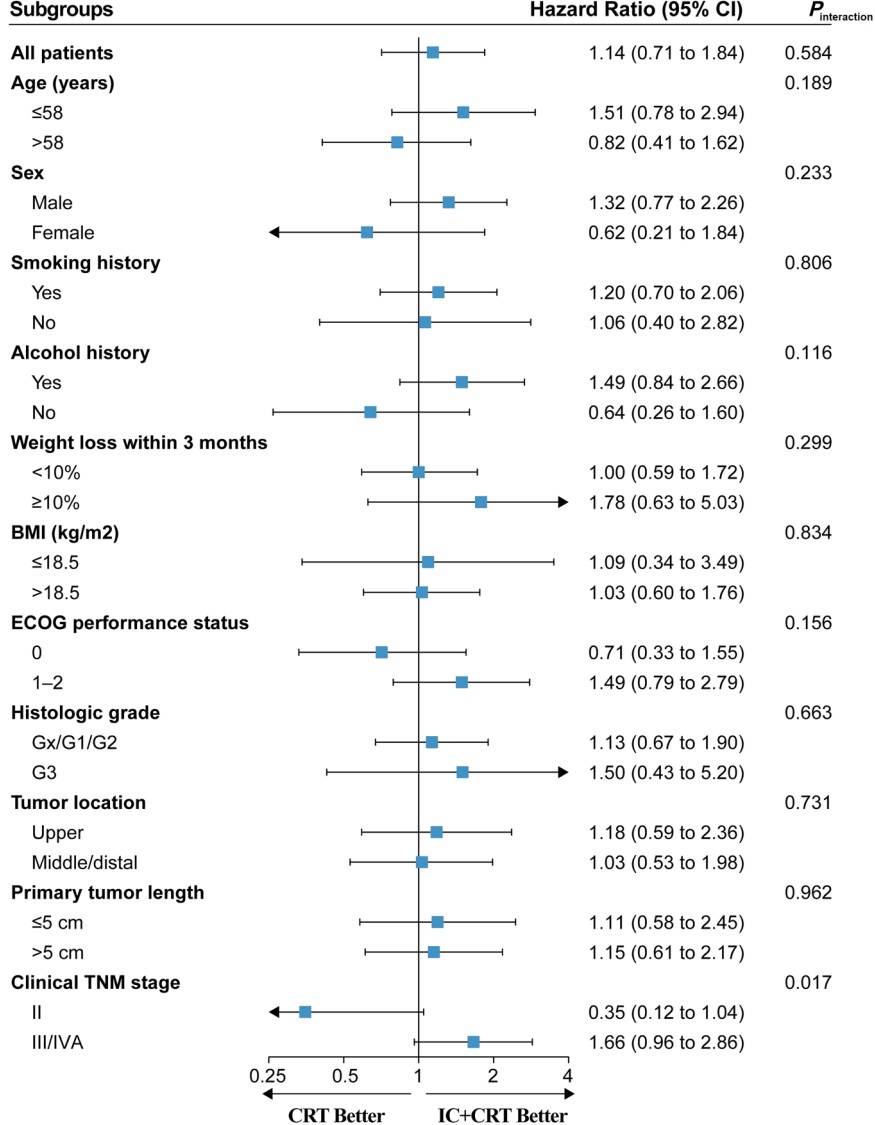

**Fig. 3 Forest plots of treatment effects on overall survival within subgroups.** Squares represent the cohort-specific hazards ratios with error bars corresponding to 95% CI bounds, which were calculated by using the univariate Cox regression model. Source data are provided as a Source Data file.

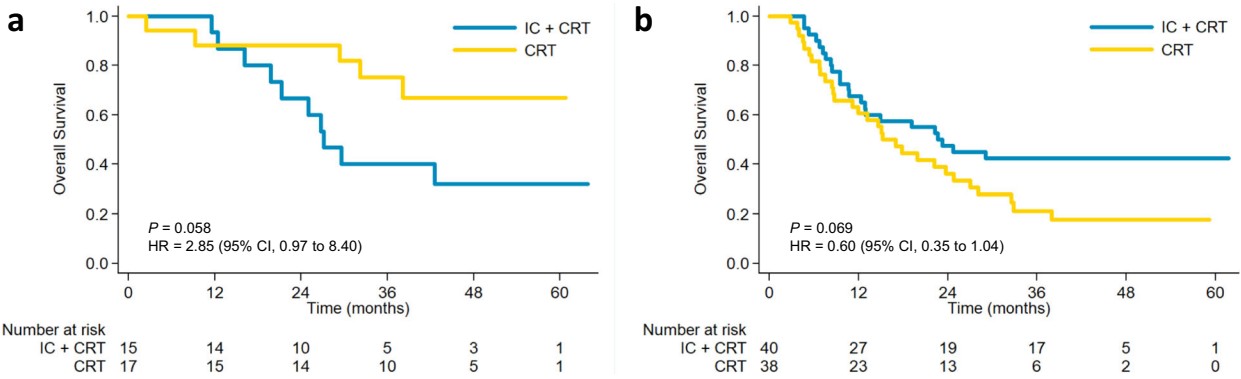

**Fig. 4 Kaplan–Meier analysis of overall survival in the two treatment groups stratified by clinical TNM stage. a** II, **b** III/IVA. Log-rank test was used (2-sided). Source data are provided as a Source Data file.

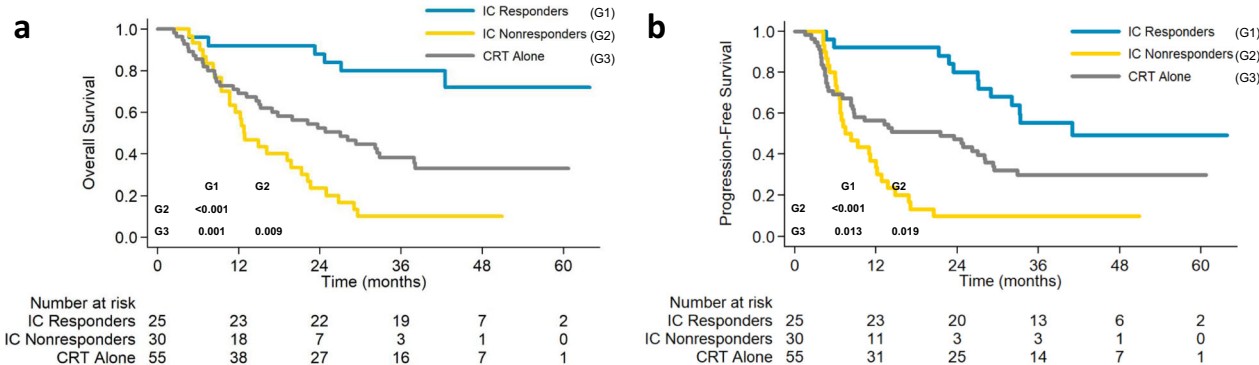

**Fig. 5 Kaplan–Meier estimates of survival curves based on the clinical response to induction chemotherapy. a** Overall survival; **b** progression-free survival. IC induction chemotherapy, CRT chemoradiotherapy. Source data are provided as a Source Data file.

**Table 3 Adverse events.**

| Event | IC + CRT group (n = 55), % | | | | CRT group (n = 55), % | | | |
|---|---|---|---|---|---|---|---|---|
| | Grade 1 or 2 | Grade 3 | Grade 4 | Grade 5 | Grade 1 or 2 | Grade 3 | Grade 4 | Grade 5 |
| Acute adverse event | | | | | | | | |
| Anemia | 53 (96.4) | 1 (1.8) | 0 | – | 50 (90.9) | 1 (1.8) | 0 | – |
| Leukopenia | 31 (56.4) | 10 (18.2) | 2 (3.6) | – | 37 (67.3) | 6 (10.9) | 4 (7.3) | – |
| Neutropenia | 18 (32.7) | 4 (7.3) | 6 (10.9) | – | 26 (47.3) | 1 (1.8) | 3 (5.5) | – |
| Febrile neutropenia | 0 | 3 (5.5) | 0 | 0 | 0 | 1 (1.8) | 0 | 0 |
| Thrombocytopenia | 22 (40.0) | 2 (3.6) | 0 | 0 | 15 (27.3) | 1 (1.8) | 1 (1.8) | 0 |
| Hepatotoxic event | 8 (14.5) | 1 (1.8) | 0 | 0 | 15 (27.3) | 2 (3.6) | 0 | 0 |
| Nephrotoxic event | 13 (23.6) | 0 | 0 | 0 | 8 (14.5) | 1 (1.8) | 0 | 0 |
| Nausea/vomiting | 53 (96.4) | 0 | 0 | 0 | 50 (90.9) | 1 (1.8) | 0 | 0 |
| Diarrhea | 6 (10.9) | 0 | 0 | 0 | 8 (14.5) | 0 | 1 (1.8) | 0 |
| Weight loss | 32 (58.2) | 0 | 0 | 0 | 28 (50.9) | 0 | 0 | 0 |
| Fatigue | 50 (90.9) | 0 | 0 | 0 | 53 (96.4) | 0 | 0 | 0 |
| Fever | 17 (30.9) | 0 | 0 | 0 | 13 (23.6) | 0 | 0 | 0 |
| Esophagitis | 35 (63.6) | 8 (14.5) | 0 | 1 (1.8) | 38 (69.1) | 13 (23.6) | 0 | 0 |
| Pneumonitis | 30 (54.5) | 2 (3.6) | 0 | 0 | 25 (45.5) | 1 (1.8) | 0 | 1 (1.8) |
| Dermatitis | 30 (54.5) | 0 | 0 | 0 | 30 (54.5) | 0 | 0 | 0 |
| Cardiac | 3 (5.5) | 0 | 0 | 0 | 4 (7.3) | 0 | 0 | 0 |
| Neurology | 3 (5.5) | 0 | 0 | 0 | 3 (5.5) | 0 | 0 | 0 |
| Late adverse event | | | | | | | | |
| Cardiac | 5 (9.1) | 0 | 0 | 0 | 4 (7.3) | 0 | 0 | 0 |
| Esophagitis | 2 (3.6) | 0 | 0 | 1 (1.8) | 3 (5.5) | 0 | 0 | 1 (1.8) |
| Pneumonitis | 13 (23.6) | 0 | 0 | 0 | 11 (20.0) | 1 (1.8) | 0 | 1 (1.8) |

A single dash (–) indicates a grade is not available.
*IC* induction chemotherapy, *CRT* chemoradiotherapy.

without statistical significance (18.2% vs. 7.3%, $P = 0.151$). No significant differences were observed in the rates of other grades 3–5 hematological adverse events or late toxicities between groups (Table 3).

## Discussion

Given the high risk of locoregional and distant recurrence after definitive CRT, more than 30% of patients with EC were treated with induction chemotherapy prior to CRT in clinical practice, despite the lack of high-level evidence[9]. Two randomized phase II trials have failed to demonstrate an obvious benefit of induction chemotherapy prior to neoadjuvant CRT and surgery in EC[16,17]. However, the role of induction chemotherapy before definitive CRT in patients with unresectable EC remains unclear. To our knowledge, this is the first randomized trial to compare the efficacy and safety of induction chemotherapy followed by definitive CRT vs. CRT alone in unresectable EC. Somewhat disappointingly, our results failed to demonstrate an obvious benefit of induction chemotherapy with docetaxel and cisplatin in the unselected ESCC population.

A number of single-arm phase II trials have explored the sequential schedule of induction chemotherapy followed by definitive CRT in EC, consistently indicating encouraging efficacy and acceptable toxicities[13–15]. Nevertheless, it is difficult to draw further conclusions from the uncontrolled phase II data. In the present controlled study, we hypothesized that the addition of induction chemotherapy to concurrent CRT could improve tumor response and survival. However, our results demonstrated that, although the induction chemotherapy group numerically had increased ORR (74.5% vs. 61.8%), the difference was not significant when compared to the CRT alone group. Moreover, the survival outcomes were very similar between the two treatment groups.

Several possible explanations can be offered for the negative results of our study. First, the sample size may not be large enough to detect the benefit of induction chemotherapy. We may have overestimated the efficacy of induction docetaxel/cisplatin, which could have affected the calculation of sample size. However, the virtually identical OS and PFS distribution in the two treatment groups suggested that even a larger sample size may not be likely to produce a significant survival advantage in favor of induction chemotherapy. Second, the current induction regimen may not be effective enough to achieve an optimal response. However, the 3-year OS rates in both groups of our trial were at least similar to or higher than the survival rates in the previous reports[3–8], suggesting docetaxel plus cisplatin is an effective regimen in ESCC. In addition, this regimen was well tolerated and did not compromise the delivery of subsequent CRT based on our data. The combination of docetaxel, cisplatin, and fluorouracil could produce a higher response rate than docetaxel plus cisplatin in EC, but the former has been demonstrated to be associated with considerable morbidity[8,18]. Thus, it is difficult to produce a very effective and safe regimen by just relying on the combination of cytotoxic drugs. Of note, recent reports revealed that the combination of chemotherapy and anti-programmed death-1 antibody provided superior ORR and survival vs. chemotherapy alone, with a manageable safety profile in advanced EC[19,20]. Whether this new combination regimen can bring more survival benefits to patients receiving definitive CRT still needs to be confirmed by prospective studies.

The third possibility is that induction chemotherapy may only benefit a certain subgroup but not unselected patients with EC. A retrospective study with 496 patients from MD Anderson Cancer Center suggested that high-risk patients (PET maximum uptake value ($SUV_{max}$) <9.7 and tumor length >5 cm or PET $SUV_{max} \geq$

9.7 and age <67 years) might obtain PFS benefits from induction chemotherapy before receiving definitive CRT[21]. In our study, the exploratory subgroup analyses found that the treatment effect was inconsistent in the clinical TNM stage subgroups (II vs. III/IVA). The benefit population of induction chemotherapy deserves further investigation.

Consistent with the results of previous studies, our exploratory, post-hoc analysis has demonstrated that tumor response after induction chemotherapy is highly predictable for outcomes in EC patients treated with CRT[22–24]. Considering the poor prognosis of the nonresponders, tumor response after induction chemotherapy could be used to guide subsequent treatment decisions, such as proceeding directly to esophagectomy or switching to an alternative chemotherapy during radiotherapy for nonresponders. Ku et al. retrospectively investigated the impact of changing chemotherapy regimen during radiotherapy in PET nonresponders after induction chemotherapy for esophageal adenocarcinoma[23]. They found that the median PFS for nonresponders who changed chemotherapy was significantly better than those who did not change chemotherapy (17.9 vs. 10.0 months; $P = 0.01$). In contrast, the same group recently reported that the PET nonresponders with ESCC who received alternative chemotherapy during radiation did not benefit from this strategy and continued to have poor outcomes, suggesting that this subsetting may have an underlying aggressive biology that cannot be countered by changing chemotherapy[24]. Therefore, the development of novel agents to overcome this unfavorable biological characteristic is much needed.

This trial has several limitations. First, this is a prospective study from a single institution. Second, due to the lack of coverage by health insurance in China, PET was recommended but not mandatory in our trial, which may have influenced the accuracy of baseline staging to some extent. Third, owing to the confounding influence of radiation-induced inflammation in the esophagus, it is difficult to assess tumor response based on CT alone. Therefore, we used the combination of CT and EGD with biopsies to evaluate the clinical response after CRT. Moreover, subsequent re-evaluation was performed if any equivocal finding was detected. Finally, we used the 6th TNM staging system in this trial, due to the limitations of the 7th TNM staging system in predicting the prognosis for EC patients treated with CRT[25].

In conclusion, compared to CRT alone, the addition of induction chemotherapy with docetaxel plus cisplatin failed to significantly improve the response rate or survival outcomes in unselected ESCC. Response to induction chemotherapy was associated with more favorable survival. It is important to investigate the benefit population of induction chemotherapy and explore more effective systemic regimens in future studies.

## Methods

**Study design**. This single-institution, open-label, randomized, phase II trial was conducted at Sun Yat-sen University Cancer Center. Eligibility criteria included the following: previously untreated, histologically proven squamous cell carcinoma of the thoracic esophagus; stage II to IVA according to the 6th TNM staging system of the American Joint Committee on Cancer; 18–70 years of age; Eastern Cooperative Oncology Group performance status of 2 or below; and adequate hematological, cardiac, pulmonary, hepatic, and renal function. Patients who had technically an unresectable disease, those considered medically unfit for surgery, or those who refused to undergo surgery were judged to be eligible. Key exclusion criteria were as follows: a history of malignancy, pregnancy or lactation, or any severe coexisting disease. For the full inclusion and exclusion criteria, refer to the research protocol in the Supplementary file. The study protocol was conducted in accordance with the Declaration of Helsinki and approved by the Institutional Review Board of Sun Yat-sen University Cancer Center and all patients provided written informed consent before enrollment. The trial is registered with ClinicalTrials.gov, number NCT02403531.

**Random assignment**. Eligible patients were randomly assigned in a 1:1 ratio in blocks of four to receive either induction chemotherapy followed by concurrent

CRT (IC + CRT group) or CRT alone (CRT group) without stratification. Random assignment was conducted by a computer-generated random number code at the Clinical Trials Center of Sun Yat-sen University Cancer Center. The random allocations were contained in sequentially numbered, opaque, sealed envelopes prepared by a statistician. Patients and clinicians were not masked to treatment assignments. After written informed consent was obtained from eligible patients, the investigators opened the envelopes sequentially and allocated patients to the corresponding treatment groups.

**Pretreatment evaluation.** Pretreatment evaluation included physical examination, routine blood workup, barium esophagogram, computed tomography (CT) with a contrast of neck, chest, and abdomen, esophagogastroduodenoscopy (EGD) with endoscopic ultrasound, pulmonary function tests, electrocardiogram, and echocardiography. Positron emission tomography (PET) was recommended but not mandatory, and bronchoscopy was optional.

**Procedures.** For patients assigned to the IC + CRT group, docetaxel (75 mg/m$^2$ on day 1) and cisplatin (75 mg/m$^2$ on day 1) were administered intravenously every 3 weeks for two cycles prior to radiotherapy. Then, radiotherapy was initiated within 21 to 28 days after the first day of the second cycle of induction chemotherapy. Docetaxel (20 mg/m$^2$) and cisplatin (25 mg/m$^2$) were administered intravenously on days 1, 8, 15, 22, and 29 concurrently with radiotherapy. Patients assigned to the CRT group only received the concurrent CRT regimen without induction treatment. Regarding radiotherapy, the recommended dose was 60.0 Gy in 28 fractions (5 days per week), administered using three-dimensional conformal radiotherapy or intensity-modulated radiotherapy (IMRT), and IMRT was preferred. Details of dose modifications and delays for chemotherapy, and radiotherapy target definition are provided in Supplementary protocol.

Hematological tests and serum biochemistry were evaluated weekly during treatment. Adverse events were recorded according to the Common Terminology Criteria for Adverse Events (version 4.0). Tumor response was assessed by physical examination, CT scan, and EGD with biopsies at 2 weeks after the second cycle of induction chemotherapy and 3 months after the completion of CRT, according to the Response Evaluation Criteria in Solid Tumors (version 1.1)[26]. Endoscopic complete response was defined as the disappearance of a tumor in the primary region without budding or ulceration, as well as negative endoscopic biopsies. The primary tumor response evaluated by CT scan was based on the vertical length and maximal transverse thickness of the tumor, as defined by Conroy et al.[5]. Two experienced imaging physicians were invited to assess the tumor response independently. If there was any equivocal finding, subsequent re-evaluation was performed within 6 weeks to determine the final response.

After CRT, patients were followed every 3 months during the first 2 years, and every 6 months thereafter. Recurrences were classified as locoregional or distant disease according to the first recurrence pattern, which were established on histologic, cytologic, or definitive radiologic evidence.

**Outcomes.** The primary endpoint was ORR, which was defined as the proportion of patients who achieved a complete response or partial response at 3 months after the completion of CRT. The secondary endpoints were OS, defined as the time from enrollment to death or censored at the last date of follow-up; progression-free survival (PFS), defined as the time from enrollment to the date of disease progression or death from any cause or censored at the last date of follow-up; adverse events; and quality of life, which will be reported in the long-term results of this study.

**Statistical analysis.** With a two-sided type I error of 0.05, a power of 80%, and a randomization ratio of 1:1, a total sample size of 98 patients would be required to demonstrate an improvement of 25% in the ORR (from 60% in the CRT group to 85% in the IC + CRT group), on the basis of the previous reports[5,14,15]. Assuming a 10% dropout rate of patients, the final sample size was 108 (54 patients per group).

Efficacy analyses were performed in the intention-to-treat population, including all patients randomly assigned to a group. Safety analyses were conducted in the safety data set, including patients who started treatment in each group. The cut-off date of data collection was September 10, 2020. Follow-up time was calculated from enrollment to the date of the last follow-up. Categorical variables were compared using the chi-square test or Fisher's exact test. Kaplan–Meier method was used to estimate OS and PFS, and log-rank test was used to examine survival differences between the two groups. We performed post-hoc subgroup analyses to assess the consistency of treatment effect on OS according to the baseline characteristics by using the univariable Cox regression model. Covariates included host factors (age, sex, smoking history, alcohol history, weight loss, body mass index, and performance status) and tumor factors (histologic grade, tumor location, primary tumor length, and clinical TNM stage). The cutoff values for age and primary tumor length were determined by the median value of the whole cohort. The cutoff for body mass index was determined as 18.5 kg/m$^2$ according to Nutritional Risk Screening 2002[27].

A competing risk analysis was used to compare the cumulative incidences of recurrences between groups. The relationship between tumor response to induction chemotherapy and outcomes was also analyzed as an exploratory, post-hoc

analysis. Statistical analyses were performed using SPSS 21.0 software (SPSS Inc., Chicago, IL) and Stata software (version 12.0), and $P < 0.05$ was considered statistically significant.

**Reporting summary.** Further information on research design is available in the Nature Research Reporting Summary linked to this article.

## Data availability

The data regarding the baseline patient information, survival outcomes, and other detailed therapeutic information have been provided as Supplementary Data 1. The remaining data are available within the Article, Supplementary Information, and Source Data. Source data are provided with this paper.

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

## Acknowledgements
We thank Prof. Jibin Li for his efforts in the statistical analysis and interpretation. This study was funded by Fundamental Research Funds for the Central Universities (19ykpy176) and Guangzhou Science and Technology Project (202102080059).

## Author contributions
M.X. and M.L. conceived and designed the trial. S.L, L.L., L.Z., Y.Z., H.L., Q.L., L.C., Y.H., B.Q., M.L., and M.X. collected clinical data. S.L, L.L., L.Z., L.Z., J.S., and Y.Y. analyzed and interpreted the data. S.L, L.L., L.Z., M.L., and M.X. accomplished the screening of the enrolled patients. All authors wrote and finally approved this paper.

## Competing interests
The authors declare no competing interests.
