## [Peer Review File · Nature Communications]

Reviewers' Comments:

Reviewer #1:

Remarks to the Author:

At present, the prognosis of inoperable esophageal cancer is not satisfactory, Induction chemotherapy followed by concurrent chemoradiotherapy as a strategy aroused clinicians' interesting for unresectable esophageal cancer. However, disputes around the value of induction chemotherapy still exist. Generally, this is not a novel study.

1. Some issues in the study design should be clarified. It is generally acceptable that the prognosis of patients with esophageal cancer after induction chemotherapy may be better. In this paper, for example, in line 28, 25 of 55 patients (45.5%) had a partial response after induction chemotherapy and were defined as responders. That means nearly half of patients in this group achieved partial response, and some of them may have opportunity to receive surgical resection (depends on the site of primary tumor). However, the design of this study did not consider surgery option for this part of patients, and it seems no one patient underwent surgical treatment later.
2. There are some key data the authors did not present sufficiently. For example: line 99 "at 3 months after CRT, 41 patients (74.5%) in the IC + CRT group versus 34 patients (61.8%) in the CRT group achieved a complete or partial response". How many patients achieved CR or PR in two groups, respectively? The specific number of CR achieved after induction chemotherapy in the IC + CRT group are very important for the result analysis.
3. Some descriptions in this study are not precise enough. For example, "to our knowledge" in line 152, this is the first randomized trial to compare the efficiency and safety of induction chemotherapy followed by definitive CRT versus CRT alone in EC ". Please read the paper published by Ajani et al (Annals of Oncology 24: 2844–2849, 2013).

Reviewer #2:

Remarks to the Author:

Thank you very much for giving me the opportunity to review the manuscript "Induction chemotherapy followed by definitive chemoradiotherapy versus chemoradiotherapy alone in esophageal squamous cell carcinoma: a randomized phase II trial" by Liu et al. In the manuscript, the results of a randomized phase II trial are presented. Primary endpoint is the overall response rate defined as complete or partial response at three months after completion of chemoradiotherapy. I reviewed the manuscript with a focus on statistical aspects. Generally, the manuscript is written well, but I have some major and some minor concerns.

Abstract:

Results on response to induction chemotherapy and on subgroup analyses might be overstated here.

Introduction:

Lines 59 - 60: The sentence "Moreover, induction chemotherapy would increase tumor response rate, which might translate into survival benefits." is not clear to me. This appears to be your research question. Please clarify or rephrase.

Results:

Confidence intervals should be added for relevant estimates, as e.g. response rates or survival probabilities.

Table 1: Cutoff values for dichotomization of age (58 years) and BMI (18.5 kg/m²) appear to be arbitrary. How and when were they determined? Were other cutoff values investigated? Distribution of continuous variables should rather be presented by means and standard deviation or medians and (interquartile) ranges.

Treatment compliance: Presentation of proportions who completed 5, 4, 3, and 2 weeks of chemotherapy (lines 87 – 89) might be confusing. It might be more intuitive to present cumulative proportions of patients who completed at least 2 weeks, at least 3 weeks, ... This also refers to Supplementary Figure 1 A.

Survival: How was median follow-up determined (see e.g. Schemper & Smith, Controlled Clinical Trials, 1996).

The first sentence in the section "Survival" (lines 104 – 106) should be rephrased. Presentation of proportions of patients who died during follow-up might be misleading due to different lengths of follow-up. Presentation of estimated survival probabilities for certain points of time as shown in lines 113 - 117 are the correct way to present the results. "After a median follow-up of ..." might also be confusing, as the presented proportions are not estimates for survival probabilities at the median follow-up time. Please quantify follow-up appropriately in a separate sentence to avoid confusion and describe the method used for quantification of follow-up time in the methods section.

For Analysis of recurrences (lines 108 – 112, Supplementary Figure 2) methods for event time analysis considering competing events should be used (see e.g. Putter, Fiocco & Geskus, Statistics in Medicine, 2007).

Exploratory Analyses: Were subgroup analyses prespecified? How and when were groups for continuous variables determined? Terms like "marginally significant trend" should be avoided. Tests on covariate-treatment interaction should be performed to test for difference in treatment effects for different patient subgroups. I would suggest adding these results to Figure 3. I also recommend using a log-scale for the hazard ratio in Figure 3 to obtain a symmetric visualization of treatment effects.

The emphasis on comparison of responders and nonresponders after induction chemotherapy to the CRT only group (lines 126 - 134) is not fully clear to me, as this cannot be influenced by the treatment decision.

Discussion: Lines 156 – 158: The same here – response to induction chemotherapy is not part of the "strategy". Please rephrase this sentence.

Lines 189 – 193: The conclusion "that only high-risk patients (...) could obtain PFS benefits from induction chemotherapy ..." based on the cited paper appears to be far too strong. Please rephrase the sentence.

Lines 193 – 197: Following your argumentation, IC + CRT was even harmful for patients with stage II ESCC in your study. Maybe you overstate results of exploratory subgroup analyses here.

Methods:

Random assignment: Random number generation (blocking, stratification, ...) and treatment allocation should be described in more detail.

Outcomes: Overall survival should not be defined "as the time from enrollment to death or the last date of follow-up", but as "time from enrollment to death censored at the last date of follow-up" or something similar. The same for progression-free survival.

Statistical analysis: "Intention-to-treat population" and "safety dataset" should be described. For time-to-event endpoints, log-rank test and Cox regression model are mentioned for group comparisons. Please clarify which method was used.

Minor comments:

- The abbreviation ORR is used on page 7 (line 127) for the first time, but it is introduced later (page 15, line 290).
- I would suggest using two decimal places for hazard ratios and corresponding confidence intervals instead of three.
- Line 292: Please change "second" to "secondary".
- Table 3: Is there a difference between "0" and "-"? Please clarify.

Reviewer #1 (Remarks to the Author):

At present, the prognosis of inoperable esophageal cancer is not satisfactory, Induction chemotherapy followed by concurrent chemoradiotherapy as a strategy aroused clinicians' interesting for unresectable esophageal cancer. However, disputes around the value of induction chemotherapy still exist. Generally, this is not a novel study.

1) Some issues in the study design should be clarified. It is generally acceptable that the prognosis of patients with esophageal cancer after induction chemotherapy may be better. In this paper, for example, in line 28, 25 of 55 patients (45.5%) had a partial response after induction chemotherapy and were defined as responders. That means nearly half of patients in this group achieved partial response, and some of them may have opportunity to receive surgical resection (depends on the site of primary tumor). However, the design of this study did not consider surgery option for this part of patients, and it seems no one patient underwent surgical treatment later.

Reply: Thanks very much for your insightful comments. Definitive chemoradiotherapy (CRT) is the standard care for patients with unresectable esophageal cancer (EC) and an alternative option to surgery for those with resectable disease. A prospective trial reported by Stahl et al found that there was no survival benefit for the addition of surgery after CRT for ESCC patients who responded to induction chemotherapy (J Clin Oncol. 2005;23(10):2310-7). In addition, there is a paucity of data regarding comparative outcomes between CRT and esophagectomy for unresectable EC patients who have the opportunity to receive curative surgery after induction chemotherapy. Therefore, we did not consider surgery as a part of patients' initial treatment strategy when we designed this trial. According to our results, the responders to induction chemotherapy showed satisfactory survival, with 3-year OS rate of 80.0%, which is more favorable compared to 50% for responders to preoperative chemotherapy treated with subsequent surgery in the RTOG 8911 trial (J Clin Oncol. 2007;25(24):3719-25). However, salvage surgery is an important option for patients who experienced locoregional recurrences after definitive CRT. We have supplemented the related information regarding salvage treatment in the revised paper (page 7, paragraph 1).

2) There are some key data the authors did not present sufficiently. For example: line 99 "at 3 months after CRT, 41 patients (74.5%) in the IC + CRT group versus 34 patients (61.8%) in the CRT group achieved a complete or partial response". How many patients achieved CR or PR in two groups, respectively? The specific number of CR achieved after induction chemotherapy in the IC + CRT group are very important for the result analysis.

Reply: Thanks for your reminding. We have clarified this information in the revised paper (page 6, paragraph 2; Table 2).

Overall, 45.5% (95% confidence interval [CI], 31.9 to 59.0) of the patients (25 of 55) had a response after induction chemotherapy and were defined as responders; no patients had a complete response, and 25 had a partial response. At 3 months after CRT, 74.5% (95% CI, 62.7 to 86.4) of the patients in the IC + CRT group (26 complete response and 15 partial response) versus 61.8% (95% CI, 48.6 to 75.1) of the patients in the CRT group (22 complete response and 12 partial response) achieved a response ($P=0.152$).

3) Some descriptions in this study are not precise enough. For example, "to our knowledge" in

line 152, this is the first randomized trial to compare the efficiency and safety of induction chemotherapy followed by definitive CRT versus CRT alone in EC ". Please read the paper published by Ajani et al (Annals of Oncology 24: 2844–2849, 2013).

Reply: Thanks for your careful comments. We have noted that two randomized phase II trials have compared the efficacy of induction chemotherapy versus no induction chemotherapy followed by neoadjuvant CRT and surgery in patients with EC (Ann Oncol, 2013; Int J Radiat Oncol Biol Phys, 2015). Different from these two trials, our study is the first randomized trial to compare the efficiency of induction chemotherapy versus no induction chemotherapy followed by definitive CRT in unresectable EC. We have added these references and clarified the differences in the revised paper (page 9, paragraph 1).

Reviewer #2 (Remarks to the Author):

Thank you very much for giving me the opportunity to review the manuscript "Induction chemotherapy followed by definitive chemoradiotherapy versus chemoradiotherapy alone in esophageal squamous cell carcinoma: a randomized phase II trial" by Liu et al. In the manuscript, the results of a randomized phase II trial are presented. Primary endpoint is the overall response rate defined as complete or partial response at three months after completion of chemoradiotherapy. I reviewed the manuscript with a focus on statistical aspects. Generally, the manuscript is written well, but I have some major and some minor concerns.

1) Abstract: Results on response to induction chemotherapy and on subgroup analyses might be overstated here.

Reply: Thanks very much for your wonderful comments. We completely agree with you and we have revised the Abstract accordingly (page 3).

2) Introduction:

Lines 59 - 60: The sentence "Moreover, induction chemotherapy would increase tumor response rate, which might translate into survival benefits." is not clear to me. This appears to be your research question. Please clarify or rephrase.

Reply: We agree that this sentence is not appropriate and we have deleted it in the revised paper (page 4, paragraph 2).

3) Results:

Confidence intervals should be added for relevant estimates, as e.g. response rates or survival probabilities.

Reply: We have added the 95% CI of response rates and OS/PFS rates in the Results section (page 6, paragraph 2; page 7, paragraph 2; page 8, paragraph 1).

4) Table 1: Cutoff values for dichotomization of age (58 years) and BMI (18.5 kg/m²) appear to be arbitrary. How and when were they determined? Were other cutoff values investigated? Distribution of continuous variables should rather be presented by means and standard deviation or medians and (interquartile) ranges.

Reply: The cutoff value for age was determined by the median value of the whole cohort. The cutoff for body mass index (BMI) was determined as 18.5 kg/m² according to Nutritional Risk Screening 2002, which is a universally accepted nutrition screening and assessment tool (Ref 27). We have clarified this information in the revised paper (page 16, paragraph 2). Also, we have reported the median values and IQR for age and BMI in Table 1.

5) Treatment compliance: Presentation of proportions who completed 5, 4, 3, and 2 weeks of chemotherapy (lines 87 – 89) might be confusing. It might be more intuitive to present cumulative proportions of patients who completed at least 2 weeks, at least 3 weeks, ... This also refers to Supplementary Figure 1 A.

Reply: Thanks very much for your reminding. We have revised this paragraph (page 5, paragraph 4) and Supplementary Figure 1A accordingly.

In the IC + CRT group, the cumulative proportions of patients who completed at least 2 weeks, 3 weeks, 4 weeks, and 5 weeks of concurrent chemotherapy during radiotherapy were 100.0%, 90.9%, 83.6%, and 65.5%, respectively. The corresponding numbers for the CRT group were 100.0%, 98.2%, 87.3%, and 80.0%, respectively (Supplementary Figure 1A).

6) Survival: How was median follow-up determined (see e.g. Schemper & Smith, Controlled Clinical Trials, 1996).

The first sentence in the section “Survival” (lines 104 – 106) should be rephrased. Presentation of proportions of patients who died during follow-up might be misleading due to different lengths of follow-up. Presentation of estimated survival probabilities for certain points of time as shown in lines 113 - 117 are the correct way to present the results. “After a median follow-up of ...” might also be confusing, as the presented proportions are not estimates for survival probabilities at the median follow-up time. Please quantify follow-up appropriately in a separate sentence to avoid confusion and describe the method used for quantification of follow-up time in the methods section.

Reply: Thanks for your comments. We have revised this section accordingly (page 6, paragraph 3). At the last follow-up on September 10, 2020, the median follow-up was 24.8 months (range, 2.4–63.9 months) for the whole cohort and 43.2 months (range, 15.4–63.9 months) for survivors. We recorded a total of 77 events of death or recurrence in the overall population, including 39 (70.9%) events in the IC + CRT group and 38 (69.1%) in the CRT group.

In addition, we have clarified the definition of follow-up time in the Methods, which was calculated from enrollment to the date of last follow-up (page 16, paragraph 2).

7) For Analysis of recurrences (lines 108 – 112, Supplementary Figure 2) methods for event time analysis considering competing events should be used (see e.g. Putter, Fiocco & Geskus, Statistics in Medicine, 2007).

Reply: According to your suggestion, we have performed a competing risk analysis to compare the cumulative incidences of recurrences between the two groups (page 16, paragraph 3). No significant differences were observed in the cumulative incidences of recurrences between the two groups ($P=0.795$) (page 6, paragraph 3; Supplementary Figure 2).

8) Exploratory Analyses: Were subgroup analyses prespecified? How and when were groups for continuous variables determined? Terms like “marginally significant trend” should be avoided. Tests

on covariate-treatment interaction should be performed to test for difference in treatment effects for different patient subgroups. I would suggest adding these results to Figure 3. I also recommend using a log-scale for the hazard ratio in Figure 3 to obtain a symmetric visualization of treatment effects.

Reply: Since the subgroups were not predefined, we have clarified that this is a post-hoc analysis in the revised paper (page 7, paragraph 3). The cutoff values for age and primary tumor length were determined by the median value of the whole cohort. The cutoff for body mass index was determined as 18.5 kg/m² according to Nutritional Risk Screening 2002²⁷. We have supplemented this information in the revised paper (page 16, paragraph 2). Also, we have deleted the inappropriate terms (page 7, paragraph 3). In addition, we performed interaction analysis and revised Figure 3 as suggested.

9) Exploratory Analyses: The emphasis on comparison of responders and nonresponders after induction chemotherapy to the CRT only group (lines 126 - 134) is not fully clear to me, as this cannot be influenced by the treatment decision.

Reply: We agree with you that the response to induction chemotherapy cannot be influenced by the treatment decision. However, previous studies demonstrated that effective induction chemotherapy had a positive effect on OS for patients with esophageal cancer who received subsequent surgery (J Clin Oncol. 2007;25:3719-3725; J Clin Oncol. 2005;23:2310-2317). Whether the response to induction chemotherapy also has a potential effect on survival outcomes in patients treated with definitive chemoradiotherapy remains unclear. Therefore, we performed this exploratory, post-hoc analysis to assess the relationship between response to induction chemotherapy and survival.

10) Discussion: Lines 156 – 158: The same here – response to induction chemotherapy is not part of the “strategy”. Please rephrase this sentence.

Reply: We have deleted this sentence in the revised paper (page 9, paragraph 1).

11) Lines 189 – 193: The conclusion “that only high-risk patients (...) could obtain PFS benefits from induction chemotherapy ...” based on the cited paper appears to be far too strong. Please rephrase the sentence.

Reply: We have revised this sentence as suggested (page 10, paragraph 2).

A retrospective study with 496 patients from MD Anderson Cancer Center suggested that high-risk patients might obtain PFS benefits from induction chemotherapy before receiving definitive CRT.

12) Lines 193 – 197: Following your argumentation, IC + CRT was even harmful for patients with stage II ESCC in your study. Maybe you overstate results of exploratory subgroup analyses here.

Reply: We agree with you that we overstated the results of subgroup analysis. We have revised this section accordingly (page 11, paragraph 1).

In our study, the exploratory subgroup analyses found that the treatment effect was inconsistent in the clinical TNM stage subgroups (II vs. III/IVA). The benefit population of induction chemotherapy deserves further investigation.

13) Methods:

Random assignment: Random number generation (blocking, stratification, ...) and treatment

allocation should be described in more detail.

Reply: We have supplemented the detailed information of random assignment in Methods (page 13, paragraph 2).

Eligible patients were randomly assigned in a 1:1 ratio in blocks of four to receive either induction chemotherapy followed by concurrent CRT (IC + CRT group) or CRT alone (CRT group) without stratification. Random assignment was conducted by a computer-generated random number code at the Clinical Trials Center of Sun Yat-sen University Cancer Center. The random allocations were contained in sequentially numbered, opaque, sealed envelopes prepared by a statistician. Patients and clinicians were not masked to treatment assignments. After written informed consent was obtained from eligible patients, the investigators opened the envelopes sequentially and allocated patients to the corresponding treatment groups.

14) Outcomes: Overall survival should not be defined “as the time from enrollment to death or the last date of follow-up”, but as “time from enrollment to death censored at the last date of follow-up” or something similar. The same for progression-free survival.

Reply: We have corrected the definitions of OS and PFS as suggested (page 15, paragraph 3).

The secondary endpoints were OS, defined as the time from enrollment to death or censored at the last date of follow-up; progression-free survival (PFS), defined as the time from enrollment to the date of disease progression or death from any cause or censored at the last date of follow-up.

15) Statistical analysis: “Intention-to-treat population” and “safety dataset” should be described. For time-to-event endpoints, log-rank test and Cox regression model are mentioned for group comparisons. Please clarify which method was used.

Reply: We have described the definition of “Intention-to-treat population” and “safety dataset” in the revised paper (page 16, paragraph 2). Efficacy analyses were performed in the intention-to-treat population, including all patients randomly assigned to a group. Safety analyses were conducted in the safety dataset, including patients who started treatment in each group.

Moreover, we have clarified the statistical method for group comparisons as suggested (page 16, paragraph 2). Kaplan-Meier method was used to estimate OS and PFS, and log-rank test was used to examine survival differences between the two groups. We performed post-hoc subgroup analyses to assess the consistency of treatment effect on OS according to the baseline characteristics by using the univariable Cox regression model.

16) Minor comments:

The abbreviation ORR is used on page 7 (line 127) for the first time, but it is introduced later (page 15, line 290).

I would suggest using two decimal places for hazard ratios and corresponding confidence intervals instead of three.

Line 292: Please change “second” to “secondary”.

Table 3: Is there a difference between “0” and “-“? Please clarify.

Reply: Thanks for your reminding. We have corrected these mistakes accordingly (page 7, paragraph 2-4; page 8; paragraph 1; page 15, paragraph 3). Regarding Table 3, we have clarified the definition of “—” in the revised paper, which indicates a grade is not available according to CTCAE.

Reviewers' Comments:

Reviewer #1:

Remarks to the Author:

Generally, the authors have addressed all my concerns. I have no any further concerns.

Reviewer #2:

Remarks to the Author:

Thank you for considering my suggestions in the revision of your manuscript. All my concerns were addressed and in my opinion the manuscript improved relevantly.

I have only two minor comments:

In the description of the interaction effects for the subgroup analyses (lines 124 to 126) you write that "interaction analysis revealed that treatment effect was consistent across all subgroups except ...". The wording implies that there is evidence for a consistent treatment effect, which is not a correct interpretation of a non-significant interaction effect. Please adapt the wording to be more precise here.

In line 134, the upper limit of the confidence interval for the overall response rate exceeds 100%. Please use an appropriate method for confidence interval estimation here, that does not produce CI limits above 100%.

Reviewer #2 (Remarks to the Author):

Thank you for considering my suggestions in the revision of your manuscript. All my concerns were addressed and in my opinion the manuscript improved relevantly. I have only two minor comments:

1) In the description of the interaction effects for the subgroup analyses (lines 124 to 126) you write that “interaction analysis revealed that treatment effect was consistent across all subgroups except ...”. The wording implies that there is evidence for a consistent treatment effect, which is not a correct interpretation of a non-significant interaction effect. Please adapt the wording to be more precise here.

Reply: Thanks very much for your comments. We completely agree with you and we have revised this sentence accordingly (page 7, paragraph 3).

The interaction analysis revealed non-significant interaction effect across all subgroups except for the clinical TNM stage subgroups ($P=0.017$).

2) In line 134, the upper limit of the confidence interval for the overall response rate exceeds 100%. Please use an appropriate method for confidence interval estimation here, that does not produce CI limits above 100%.

Reply: Thanks for your reminding. We have corrected the confidence interval in the revised paper (page 8, paragraph 1).